# Phenolics and Carotenoid Contents in the Leaves of Different Organic and Conventional Raspberry (*Rubus idaeus* L.) Cultivars and Their In Vitro Activity

**DOI:** 10.3390/antiox8100458

**Published:** 2019-10-07

**Authors:** Alicja Ponder, Ewelina Hallmann

**Affiliations:** Department of Functional and Organic Food, Institute of Human Nutrition, Warsaw University of Life Sciences, Nowoursynowska 159c, 02-776 Warsaw, Poland; alicja_ponder@sggw.pl

**Keywords:** raspberry leaves, phenolics, carotenoids, antioxidant activity, organic, conventional

## Abstract

Raspberry leaves are a source of carotenoids and polyphenols, including ellagic acid and salicylic acid. The results of scientific research suggest that they have potential pro-health properties that contribute to human health. The aim of this study was to determine the polyphenolic and carotenoid profiles in the leaves of selected raspberry cultivars and their in vitro activity. The second aim was to determine the impact of organic and conventional farm management on the polyphenol, carotenoid, and chlorophyll contents in different raspberry cultivars: ‘Polana’, ‘Polka’, ‘Tulameen’, ‘Laszka’ and ‘Glen Ample’. Compared with conventional raspberry leaves, organic raspberry leaves were characterized by a significantly higher content of dry matter, total polyphenols, total phenolic acids, chlorogenic acid, caffeic acid, salicylic acid and quercetin-3-*O*-rutinoside; moreover, the organic leaves were characterized by higher antioxidant activity. Among examined cultivars, ‘Polka’ c. was characterized by the highest antioxidant status. However, raspberry leaves from conventional farms contained more total carotenoids, violaxanthin, alpha-carotene, beta-carotene, total chlorophyll and individual forms of chlorophylls: a and b.

## 1. Introduction

Raspberry (*Rubus ideaus*) is recognized by consumers as a tasty and healthy fruit. Recent research indicates that the leaves of berry plants, such as strawberries, raspberries, blueberries and blackcurrants, are a potential source of bioactive compounds with strong, pro-health, anticancer and anti-inflammatory properties [1,2,3,4]. Berry leaves are by-products of berry plant cultivation. Their traditional therapeutic use against several diseases, such as the common cold, inflammation, diabetes, and ocular inflammation, has been almost forgotten [5]. Raspberry leaves contain high amounts of polyphenols and can serve as a potential source of natural antioxidants for medicinal and commercial uses. Raspberry leaves contain phenolic acids, such as chlorogenic, gallic, ferulic, caffeic acids, as well as flavonoids, including quercetin and kaempferol-3-*O*-glucosiden [6]. However, two chemical compounds deserve special attention: ellagic and salicylic acids [7,8]. These compounds show strong biological effects in vitro that have been connected to pharmacological and nutritional effects [9]. They are mainly related to the prevention of cardiovascular diseases [10]. Many current medicines are derived from plants, including aspirin, which is a synthetic derivative of salicylic acid [11,12]. Plants produce salicylic acid as a response to biotic (pest and diseases) stresses [13,14].

Many studies have shown that organic cultivation methods increase the amount of bioactive compounds in fruits [15,16,17], mostly due to the effect of plant self-protection against pest and diseases. The latest research with raspberry fruit indicates that organic fruit contained significantly more polyphenols compare to conventional one fruits [18]. Phenolic compounds are well known as “natural pesticides”. In organic agriculture, plants produce more phenolic compounds in the leaves because use of synthetic pesticides is forbidden [19,20]. In the present experiment, the authors want to investigate how cultivation methods effect the content of biologically active compounds in raspberry leaves. 

The main goal of the study was to determine the polyphenol, carotenoid and chlorophyll profile in leaves of selected raspberry cultivars. The second aim was to determine the impact of farm management on the bioactive compound contents in different raspberry leaves cultivars. 

## 2. Materials and Methods 

### 2.1. Plants Origin

The experiment was carried out in 2013. Leaves of five raspberry cultivars (‘Polana’, ‘Polka’, ‘Tulameen’, Laszka’ and ‘Glen Ample’) were collected at the time of cultivation. Three organic and neighborhood conventional farms were used for experimental purposes. From one cultivar (one field plot), 3–4 plants were chosen, which were analyzed separately. One sample consisted of 10 leaves. The farm was treated as a replication. Detailed information about the agricultural conditions and practices in experimental farms is presented in Table 1 and Figure 1.

### 2.2. Plant Material Preparation

The leaves for chemical analysis were harvested early in the morning from each production farm and immediately transported to the laboratory. Each sample was divided into two parts. The first part was used for dry matter evaluation, and the second part was freeze-dried using a Labconco (2.5) freeze-dryer (Warsaw, Poland, −40 °C, pressure 0.100 mBa). After freeze-drying, the plant material was ground in a laboratory mill (A-11). The ground samples were then stored at −80 °C.

### 2.3. Dry Matter Content

The dry matter content of the raspberry leaves was measured before freeze-drying. The dry matter content was determined using the weight method. Empty glass beakers were weighed, filled with fresh leaves and weighed again. The samples were dried at 105 °C for 72 h in an FP-25W Farma Play (Tczew, Poland) dryer. After 3 days, the samples were cooled to 21 °C and weighed. The dry matter content was calculated for the leaf samples based on their mass differences and given in units of 100 g^−1^ FW (fresh weight) [21].

### 2.4. Phenolic Acid and Flavonol Separation and Identification

Polyphenols were measured by an HPLC (High Performance Liquid Chromatography) method that was described previously in detail by Hallmann et al. [22]. A total of 100 mg of freeze-dried, powdered plant material was mixed with 5 mL of 80% methanol and shaken on a Micro-Shaker 326 M (Marki, Poland). Next, all samples were extracted in an ultrasonic bath (10 min, 30 °C, 5500 Hz). After 10 min of extraction, the leaf samples were centrifuged (10 min, 3780× *g*, 5 °C). The supernatant was collected in a clean plastic tube and centrifuged again (5 min; 31,180× *g*, 0 °C). A total of 900 µL of supernatant was transferred to an HPLC vial and analysed. For polyphenol compound separation and identification, a Synergi Fusion-RP 80i Phenomenex column (250 × 4.60 mm) was used. The analysis was carried out with the use of Shimadzu equipment (USA Manufacturing Inc, Lebanon, IN, USA: two pumps LC-20AD, controller CBM-20A, column oven SIL-20AC, spectrometer UV/Vis SPD-20 AV). The phenolic compounds were separated under gradient conditions with a flow rate of 1 mL min^−1^. Two gradient phases were used: 10% (*V:V*) acetonitrile and ultrapure water (phase A) and 55% (*V:V*) acetonitrile and ultrapure water (phase B). The phases were acidified by orthophosphoric acid (pH 3.0). The total time of the analysis was 38 min. The phase-time programme was as follows: 1.00–22.99 min, 95% phase A and 5% phase B; 23.00–27.99 min, 50% phase A and 50% phase B; 28.00–28.99 min, 80% phase A and 20% phase B; and 29.00–38.00 min, 95% phase A and 5% phase B. The wavelengths were 250 nm for flavonols and 370 nm for phenolic acids. The phenolic compounds were identified by using 99.9% pure standards (Sigma-Aldrich, Szelągowska, Poland) and the analysis times for the standards (Figure 2 and Figure 3).

### 2.5. Carotenoid and Chlorophyll Separation and Identification

Carotenoids and chlorophylls were measured by an HPLC method, as described by Hallmann et al. [22]. Sample preparation included extraction of 100 mg of freeze-dried sample with 100% acetone using an ultrasonic cold bath (10 min, 0 °C). Samples were then centrifuged (10 min, 3780× *g*, 0 °C). One millilitre of supernatant was transferred into an HPLC vial. The HPLC setup used to determine carotenoids and chlorophylls consisted of two LC-20AD pumps, a CMB-20A system controller, an SIL-20AC autosampler, an ultraviolet–visible SPD-20AV detector, a CTD-20AC oven and a Max-RP 80A column (250 × 4.60 mm), which are all Shimadzu products (Polish agent Shimpol, Warsaw, Poland). Methanol + acetonitrile (phase A) and methanol + ethyl acetate (phase B) at a flow rate of 1 mL min^−1^ were used as the gradient solvents (1.00–14.99 min, 100% phase A, 15.00–22.99 min, 40% phase A; and 24.00–27.00 min, 100% phase A). The wavelength used for detection was 445–450 nm. The carotenoid and chlorophyll concentrations were calculated using standard curves and the sample dilution coefficient and presented in mg per 100 g of fresh material. Identified carotenoids and chlorophylls are presented in Figure 4. 

### 2.6. ABTS^·+^ Radical Cation Scavenging Activity Assay

#### 2.6.1. ABTS Reagent Preparation

Twenty milliliters of distilled water was added to 0.0265 g of potassium persulfate (K_2_S_2_O_8_). Five milliliters of distilled water followed by 5 mL of a previously prepared aqueous solution of potassium persulfate was added to 0.0384 g of ABTS^·+^ (2’2-azinebis-3-ethylbenzothiazolin-6-sulfonic acid) reagent. The solution was prepared a minimum of 12 h before the planned assay and stored in a dark place.

#### 2.6.2. Antioxidant Activity Measurement

Two-hundred and fifty milligrammes of the sample of freeze-dried plant material tested was weighed into a sterile falcon tube plastic tube with a cap (50 mL), and 25 mL of distilled water was added. It was placed onto a vortex shaker (LP shaker Vortex, Labo Plus, Warsaw, Poland) for 60 s at 2000 rpm, for complete mixing. Subsequently, the sample was incubated in a shaker incubator (IKA KS 4000 Control, IKA, Staufen im Breisgau, Germany) for 60 min (temperature 30 °C, 6× *g*). After incubation, the sample was again shaken on a vortex shaker for 60 s for complete mixing and then centrifuged (Centrifuge, MPW-380 R, Warsaw, Poland) at 5 °C and 14,560× *g* for 20 min. After centrifugation, the supernatant was used for determinations. In 10 mL glass tubes, test extract solution, measured with a predetermined dilution scheme (0.5–1.5 mL), was then added to 3.0 mL of ABTS^·+^ cationic solution in PBS (phosphate-buffered saline). Absorbance measurements were taken exactly 6 min after incubation at room temperature. Absorbance was measured at a wavelength λ = 734 nm using a spectrophotometer (Helios γ, Thermo Scientific, Warsaw, Poland). The obtained measurements were calculated using special formula including the dilution factor. The final results were express as mmol of TE (Trolox equivalents per 100 g FW (fresh weight of leaves)) [23]. 

### 2.7. Statistical Analysis

The results obtained from the chemical analyses were statistically analyzed using Statgraphics Centurion 15.2.11.0 software (StatPoint Technologies, Inc., Warranton, VA, USA). The values presented in the tables are expressed as the mean values for the organic and conventional cultivation systems for the five raspberry cultivars (‘Polana’, ‘Polka’, ‘Tulameen’, ‘Laszka’ and ‘Glen Ample’). The mean value for the organic raspberry leaves was obtained from 20 individual measurements (*n* = 20), and the mean value for conventional raspberry leaves was obtained from 24 measurements (*n* = 24). Individual raspberry cultivars were represented as follows: ‘Polana’ (*n* = 8); ‘Tulameen’ (*n* = 8); ‘Laszka’ (*n* = 8); ‘Glen Ample’ (*n* = 8); ‘Polka’ (*n* = 12). The statistical calculations were based on two-way analysis of variance with the use of Tukey’s test (*p* = 0.05). A lack of statistically significant differences between the examined groups is indicated by similar letters. The standard error (SE) is provided with each mean value reported in the tables.

## 3. Results 

### 3.1. Polyphenol Content

The dry matter and polyphenol contents in examined raspberry leaves are presented in Table 2. Organic raspberry leaves were characterized by a significantly higher content of dry matter (*p* = 0.0055) and total polyphenols (*p* = 0.0001), including total phenolic acids (*p* < 0.0001) as well individual acids: chlorogenic, caffeic and salicylic acids. For flavonoids, we observed that organic plants, compared with conventional plants, contained significantly more quercetin-3-*O*-rutinoside (*p* = 0.0009). However, raspberry leaves from conventional farming systems contained significantly more luteolin (*p* = 0.0117) than did leaves from organic farming systems. 

Raspberry cultivar had a significant impact on the content of phenolic compounds in examined leaves (Table 3). ‘Tulameen’ cv. was characterized by the highest level of caffeic acid and quercetin derivates, whereas ‘Polka’ cv. contained the highest and significant levels of ellagic acid (p = 0.0046). Both of these cultivars contained significantly more quercetin-3-*O*-glycoside than did the other examined cultivars. The highest luteolin content was found in the leaves of raspberry cultivars ‘Polka’ and ‘Glen Ample’. However, the highest content of quercetin among all analysed cultivars was found in the leaves of raspberry ‘Polana’ cv. 

### 3.2. Carotenoid and Chlorophyll Contents

The contents of carotenoids and chlorophylls are presented in Table 4. The results showed that raspberry leaves from conventional farming contained significantly more total carotenoids (*p* = 0.0014), violaxanthin (0.026 mg 100 g^−1^ FW and 0.017 mg 100 g^−1^ FW), alpha-carotene (0.109 mg 100 g^−1^ FW and 0.060 mg 100 g^−1^ FW) and beta-carotene (1.22 mg 100 g^−1^ FW and 0.46 mg 100 g^−1^ FW) than did the leaves from organic farming; however, the leaves from conventional farming contained significantly less neoxanthin (*p* = 0.003), lutein (*p* = 0.0069) and zeaxanthin (*p* = 0.0118). Moreover, leaves from conventional farming, compared to leaves from organic farming, were characterized by higher contents of total chlorophylls (10.52 mg 100 g^−1^ FW and 5.75 mg 100 g^−1^ FW) and individual forms of chlorophyll (a and b). Cultivar had a significant impact only on neoxanthin (*p* < 0.0001) content in leaves. ‘Laszka’ cv. contained significantly more of this xanthophyll among all analysed cultivars (Table 5).

### 3.3. Antioxidant Activity

Organic raspberry leaves, compared with leaves from conventional farming, were characterized by significantly higher antioxidant activity (*p* < 0.0001) (Figure 5). Among the group of examined raspberry cultivars, the strongest antioxidant potential shown was in ‘Polka’ cv. and ‘Tulameen’ cv. There was a significant correlation between antioxidant activity in vitro and the total polyphenol content in examined raspberry leaves (Figure 6). The stronger antioxidant activity in the leaves was a reflection of higher content of polyphenols, especially in the organic plants (R^2^ = 0.8302, *p* < 0.0001).

## 4. Discussion 

Berry fruits are recognized worldwide as “superfoods” due to the high content of bioactive compounds and their health benefits [24,25,26,27]. Most research on the impact of the cultivation system (organic and conventional) on the quality of raspberries concerns fruit [15,18]. As the literature shows, only little attention has been paid to biologically active substances in leaves, which was the research material in this study. Their traditional therapeutic use against several diseases, such as the cold, inflammation, diabetes and ocular dysfunction, has almost been forgotten. Raspberry leaves are a powerful source of biologically active compounds (Table 2, Table 3, Table 4 and Table 5). The high content of bioactive compounds means that infusions of raspberry leaves can be used in phytotherapy [5]. Only a few studies have shown the antioxidant properties and polyphenol content in raspberry leaves. In our analysis, we found that organic raspberry leaves, compared with conventional raspberry leaves, were characterized by significantly higher contents of total polyphenols, total phenolic acids, chlorogenic acid, caffeic acid, salicylic acid and quercetin-3-*O*-rutinoside. In our experiment, we found 136.1 mg 100 g^−1^ FW of total polyphenols in organic leaves and 119.9 mg 100 g^−1^ FW of total polyphenols in conventional leaves. Teleszko & Wojdyło [28] described similar results. Their research showed that the leaves of berries were not only a valuable source of antioxidants but also contained significantly more polyphenols than did the fruit. This clearly indicates that plant parts other than the fruits could be used for medical or food purposes. An example of their application is herbal tea. The main compound of polyphenols found in raspberry leaves is salicylic acid. We found 3.51 mg 100 g^−1^ FW of salicylic acid in the studied organic raspberry leaves. Salicylic acid is synthesized by plants as a response to abiotic stresses, such as osmotic stress, chilling, drought and heat [13,14,29] As reported by Nour, Trandafir & Cosmulescu [9], seven cultivars of blackcurrant contain salicylic acid in the leaves, ranging from 3.97 mg 100 g ^−1^ FW to 5.28 mg 100 g ^−1^ FW; in this study, raspberry leaves contained 2.51 mg 100 g ^−1^ FW to 3.20 mg 100 g ^−1^ FW. From a chemical point of view, salicylic acid could be used as a substrate for acetylic reactions and acetylsalicylic acid formation. The manifold effects of acetyl salicylic acid on human physiology can potentially provide health benefits [30]. One of the important phenolic acids extracted from the *Rubus* family is ellagic acid. As reported by Landete [10] raspberry fruits contain ellagic acid in a wide rate: 47–270 mg 100 g^−1^ FW. In a study by Oszmiański et al. [1] the ellagic acid content in raspberry leaves ranged from 215.5 mg 100 g^−1^ FW to 1078.5 mg 100 g^−1^ FW. In our experiment, we found much lower levels of ellagic acid but were still satisfied (Table 2 and Table 3). 

A very important group of compounds present in raspberry leaves is flavonoids. The quantity of flavonoids in the leaves of raspberries is significantly higher than that in the fruits, where flavonoids compose only a very small fraction of the bioactive compounds [31]. In a study by Oszmiański et al. [1] the flavonoid fraction was the main phenolic group, constituting almost 11% of leaf extract powder weight. In our experiment, we identified 5 flavonoid compounds, quercetin-3-*O*-rutinoside, quercetin-3-*O*-glucoside, luteolin, myricetin and quercetin, in examined raspberry leaves. However, Buricova et al. [32] examined antioxidant capacity and antioxidants in raspberry leaf water extract, and three flavonoid compounds were detected (catechin, epicatechin and procyanidin B_1_), while Ferlemi & Lamari [5] detected much more flavonoid compounds present in raspberry leaves (quercetin, quercetin-3-*O*-rutinoside, quercetin-3-*O*-galactoside, quercetin-3-*O*-glucoside, quercetin-3-*O*-glucuronide, kaempferol-3-*O*-glucoside, epicatechin gallate methyl gallate, sanguiin H-6/lambertianin C and lambertianin D). In contrast, in an analysis by Oszmiański et al. [1] thirty-three phenolic compounds were detected in wild blackberry leaf samples (fifteen flavonols, thirteen hydroxycinnamic acids, three ellagic acid derivatives and two flavones). Flavonoids have antioxidant abilities and protect plants from various biotic and abiotic stresses. The role of secondary metabolic pathways in plant responses is to cope with oxidative stress, resulting in the synthesis of flavonoids [33]. Another important role of flavonoids in foliar plants is their action as a screen against severe sunlight illumination [34]. One of the most important priorities in research on polyphenolic compound content is not just determining their presence but also biological activity in vitro and in vivo. Dudzińska et al. [35] investigated the polyphenol content in raspberry leaves and their antioxidative power. The antioxidant capacities of the examined extracts remained relatively high and corresponded well to the determined total polyphenol content. As pointed out by Oszmiański et al. [1] the antioxidant power of raspberry leaves is strongly connected with the total polyphenol content. They measured total phenolic content and antioxidant activity (AA) of 27 species belonging to the *Rubus* family. They found a significant link between the highest polyphenolic concentration and AA of raspberry leaves. The species with the highest total polyphenol content also had the highest antioxidant activity: *Rubus pedemontanus* (205 mol TE g^−1^ DW (dry weight) and 310.88 mg 100 g^−1^ DW of polyphenols) and *Rubus partenocissus* (203 mol TE g^−1^ DW and 298.74 mg 100 g^−1^ DW of polyphenols); species with the lowest value were characterized by the lowest antioxidant power: *Rubus radula* (151 mol TE g^−1^ DW and 202.21 mg 100 g^−1^ DW of polyphenols) and *Rubus nesseris* (91 mol TE g^−1^ DW and 85.51 mg 100 g^−1^ DW of polyphenols). In our study, we observed the highest levels of total polyphenols in ‘Polka’ cv. (151.75 mg 100 g^−1^ FW) and ‘Tulameen’ cv. (136. 95 mg 100 g^−1^ FW). Those levels were reflected in their antioxidant status (88.10 mmol Trolox 100 g^−1^ FW and 80.22 mmol Trolox 100 g^−1^ FW) and significant correlation between features (R^2^ = 0.8302, *p* < 0.0001) for the organic raspberry but much weaker correlation for the conventional raspberry (R^2^ = 0.6227, *p* < 0.0001) (Figure 6). Similar results were described by Zlotek et al. with basil leaves [36]. The antioxidant status of leaves was positively correlated with polyphenols content [37]. 

In addition to the presence of photosynthetic pigments, carotenoids also exist in raspberry leaves. Their concentration in leaves depends mainly on the level of chlorophyll. The higher the concentration of chlorophyll in the leaves, the more carotenoids present. Chlorophyll is associated with the function of carotenoids, which are produced by plants mainly to protect the photosynthetic system against photooxidation. Carotenoids are synthesized via the general biosynthetic pathway within the chloroplasts of plants. Shen et al. [38] also studied the effect of increased UV-B radiation on carotenoid accumulation and total antioxidant capacity in tobacco (*Nicotiana tabacum* L.) leaves. Higher levels of chlorophylls were positively correlated with beta-carotene content. In our experiment, we observed similar results. Conventional raspberry with a significant chlorophyll level (2.43 mg 100 g^−1^ FW) contained a significant level of total carotenoids (3.14 mg 100 g^−1^ FW) compared to that of organic raspberry (1.79 mg 100 g^−1^ FW and 2.61 mg 100 g^−1^ FW, respectively) (Table 4). 

## 5. Conclusions

In summary, the aim of the study was reached and confirmed. Raspberry leaves are a valuable source of bioactive compounds. Moreover, compared to conventional leaves, organic raspberry leaves were characterized by a significantly higher content of total polyphenols, total phenolic acids, chlorogenic acid, caffeic acid, salicylic acid and quercetin-3-*O*-rutinoside. Additionally, the organic leaves had higher antioxidant activity; the strongest antioxidant potential was shown by the ‘Polka’ and ‘Tulameen’ cultivars. On the other hand, raspberry leaves from conventional farms contained more total carotenoids, violaxanthin, alpha-carotene, beta-carotene, total chlorophyll and individual forms of chlorophyll (a and b). The mineral fertilization used in conventional agriculture increases the level of these compounds.

## Figures and Tables

**Figure 1 antioxidants-08-00458-f001:**
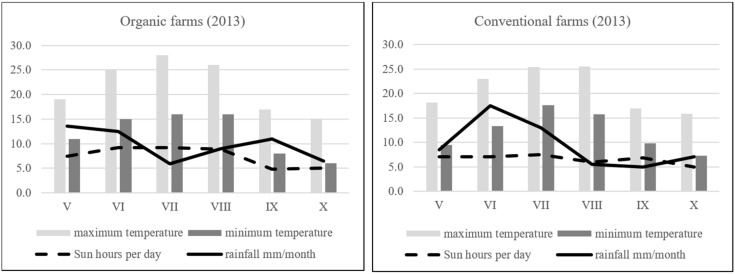
Weather conditions at the experimental farms (organic and conventional) in time of raspberry leaves experiment.

**Figure 2 antioxidants-08-00458-f002:**
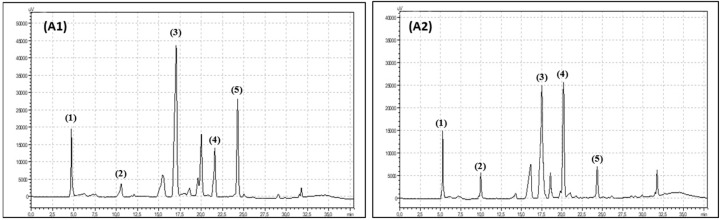
Chromatogram showing retention times for organic raspberry (**A1**) and conventional raspberry leaves (**A2**) phenolic acids: (1) chlorogenic acid, (2) caffeic acid, (3) salicylic acid, (4) p-coumaric acid, (5) ellagic acid.

**Figure 3 antioxidants-08-00458-f003:**
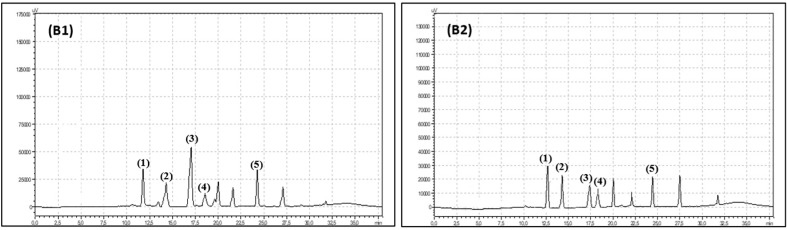
Chromatogram showing retention times for organic raspberry (**B1**) and conventional raspberry leaves (**B2**) flavonoids: (1) quercetin-3-*O*-rutinoside, (2) myrycetin, (3) quercetin-3-*O*-glucoside, (4) quercetin, (5) luteolin.

**Figure 4 antioxidants-08-00458-f004:**
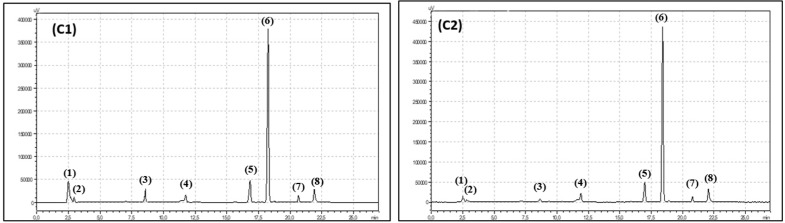
Chromatogram showing retention times for organic raspberry (**C1**) and conventional raspberry leaves (**C2**) carotenoids and chllorophylls: (1) lutein, (2) zeaxanthin, (3) neoxanthin, (4) violaxanthin, (5) chlorophyll b, (6) chlorophyll a, (7) alpha-carotene, (8) beta-carotene.

**Figure 5 antioxidants-08-00458-f005:**
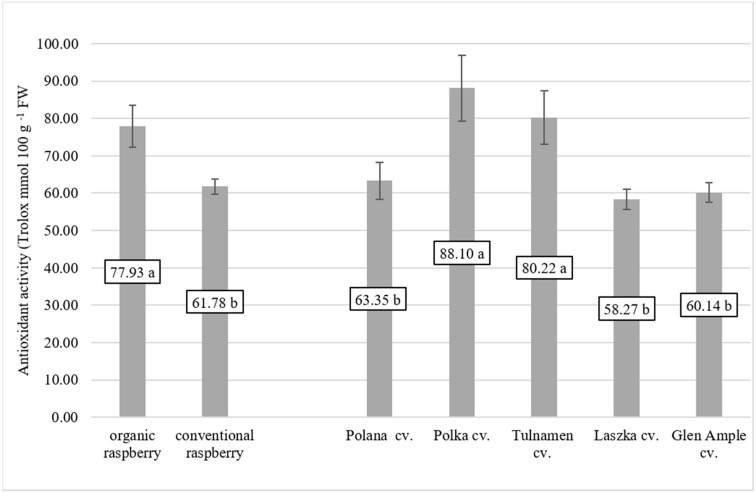
Antioxidant activity raspberry from organic and conventional cultivation (*p* < 0.0001) and raspberry cultivars (*p* < 0.0001). Means followed by the same letter are not significantly different (*p* < 0.05).

**Figure 6 antioxidants-08-00458-f006:**
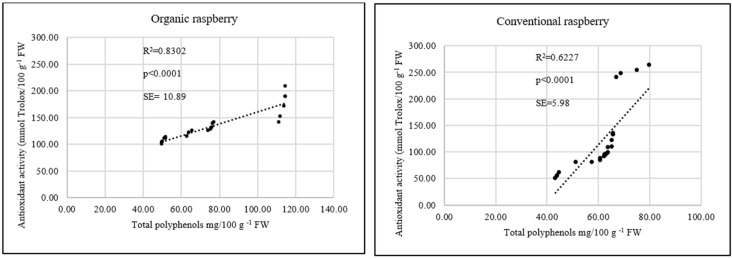
Linear regression (Pearson’s coefficient R^2^) between antioxidant activity and total polyphenols in organic (*n* = 20) and conventional (*n* = 24) raspberry leaves.

**Table 1 antioxidants-08-00458-t001:** Characterization of localization, fertilizers regime and plant protection used for organic and conventional raspberry. Cultivation in 2013 year.

Cultivation System	Localization	Type of Soil	Kind of Fertilizer	Dose of Fertilizers and Time of Given	Plant Protection System
organic farm no. 1	Zakroczym	sandy middle soil IVa and IVb category (15% floatable particles) pH 5.5	cow manure	35 t ha^−1^ one year before raspberry planting	Grevit 200 SL
	(52°26′′ N 20°36′′ E)				
organic farm no. 2	Załuski	sandy middle soil, sandy-clay IV category (20% floatable particles), pH 5.5	cow manure	30 t ha^−1^ one year before raspberry planting	no protection
	(52°37′′ N 20°22′′ E)				
organic farm no. 3	Radzanów	sandy middle soil IVa and III category (10% floatable particles), pH 6.0	sheep manure, green manure	10 t ha^−1^ and 15 t ha^−1^ one year before raspberry planting,	Bioczos 33 SL, Grevit 200 SL
	(51°33′′ N 20°51′′ E)				
conventional farm no. 1	Czerwińsk nad Wisłą	sandy-loamy middle soil IV and III category (20% floatable particles), pH 5.5	Hydrocomplex 12-11-18; Superba 8-11-36	(200 kg ha^−1^, 150 kg ha^−1^) in autumn a year before raspberry planting; 3 doses in time of cultivation	Signum 33 WG, Miros 20 SP,
	(52°23′′ N 20°20′′ E)				
conventional farm no. 2	Czerwińsk nad Wisłą	sandy-loamy middle soil IV and III category (25% floatable particles), pH 5.5	amonium nitrate, polyphosphate, magnesium sulphate	in autumn a year before raspberry planting; 3 doses in time of cultivation	Calypso 480 SC, Miros 20 SP, Zato 50 WG
	(52°23′′ N 20°20′′ E)				
conventional farm no. 3	Czerwińsk nad Wisłą	sandy-clay middle soil II and III category (20% floatable particles) pH 6.0	Rosafert 5-12-24-3	250 kg ha^−1^ in autumn a year before raspberry planting; 4 doses in time of cultivation	Calypso 480 SC, Miros 20 SP, Zato 50 WG
	(52°25′′ N 20°23′′ E)				

**Table 2 antioxidants-08-00458-t002:** The content of dry matter in (g 100 g^−1^ FW) and polyphenols (mg 100 g^−1^ FW) in examined raspberry leaves depending on cultivation system. Data are presented as the mean ± SE with ANOVA *p*-value.

Examined Compounds	Organic Raspberry (*n* = 20)	Conventional Raspberry (*n* = 24)	*p*-Value
dry matter	29.81 ± 1.26a ^1^	25.64 ± 0.73b	0.0055
total polyphenols	136.10 ± 6.86a	119.95 ± 14.19b	0.0001
total phenolic acids	64.09 ± 3.80a	52.94 ± 6.42b	< 0.0001
chlorogenic	5.66 ± 0.57a	3.81 ± 0.50b	0.0188
caffeic	24.98 ± 3.32a	4.64 ± 0.82b	< 0.0001
p-coumaric	14.77 ± 0.83b	25.02 ± 3.89a	0.0243
ellagic	15.18 ± 2.41a	16.96 ± 2.46a	N.S. ^2^
salicylic	3.51 ± 1.12a	2.51 ± 0.08b	< 0.0001
total flavonoids	72.01 ± 3.77a	67.01 ± 7.47a	N.S.
quercetin-3-*O*-rutinoside	5.40 ± 1.12a	1.42 ± 0.34b	0.0009
quercetin-3-*O*-glucoside	31.31 ± 4.13a	21.49 ± 3.55a	N.S.
luteolin	8.87 ± 1.43b	20.94 ± 3.90a	0.0117
myrycetin	7.84 ± 1.10a	6.33 ± 2.78a	N.S.
quercetin	18.59 ± 1.12a	16.84 ± 2.78a	N.S.

^1^ Means in rows followed by the same letter are not significantly different at the 5% level of probability (*p* < 0.05); ^2^ N.S. not significant statistically.

**Table 3 antioxidants-08-00458-t003:** The content of dry matter in (g 100 g^−1^ FW) and polyphenols (mg 100 g^−1^ FW) in examined raspberry leaves depending on cultivar. Data are presented as the mean ± SE with ANOVA *p*-value.

Examined Compounds	‘Polana’ cv. (*n* = 8)	‘Polka’ cv. (*n* = 12)	‘Tulameen’ cv. (*n* = 8)	‘Laszka’ cv. (*n* = 8)	‘Glen Ample’ cv. (*n* = 8)	*p*-Value
dry matter	26.06 ± 0.73a ^1^	27.77 ± 1.06a	29.72 ± 2.88a	28.85 ± 1.86a	25.14 ± 0.81a	N.S. ^2^
total polyphenols	128.51 ± 2.78a	151.75 ± 20.67a	136.95 ± 19.10a	88.08 ± 13.17a	118.95 ± 7.48	N.S.
total phenolic acids	62.81 ± 1.83a	66.55 ± 8.594a	55.55 ± 11.30a	37.35 ± 7.16a	63.52 ± 4.36a	N.S.
chlorogenic	4.46 ± 0.59a	5.13 ± 0.83a	4.96 ± 0.17a	6.04 ± 1.40a	2.44 ± 0.12a	N.S.
caffeic	6.56 ± 1.59a	12.60 ± 1.40ab	27.00 ± 8.70b	8.61 ± 3.59ab	15.27 ± 4.99ab	0.0401
p-coumaric	28.81 ± 6.15a	22.67 ± 5.92a	13.92 ± 1.17a	10.10 ± 2.43a	25.14 ± 2.24a	N.S.
ellagic	20.04 ± 5.10ab	23.22 ± 3.89b	6.46 ± 1.01a	9.96 ± 1.50a	17.54 ± 0.87ab	0.0046
salicylic	2.93 ± 0.11a	2.92 ± 0.36a	3.20 ± 0.36a	2.65 ± 0.27a	3.13 ± 0.23a	N.S.
total flavonoids	65.71 ± 2.62a	85.19 ± 12.43a	81.40 ± 7.84a	50.73 ± 6.37a	55.43 ± 3.14a	N.S.
quercetin-3-*O*-rutinoside	2.12 ± 0.17ab	4.95 ± 1.01ab	7.05 ± 2.22b	0.53 ± 0.18a	0.63 ± 0.18a	0.0010
quercetin-3-*O*-glucoside	14.80 ± 1.10a	36.43 ± 4.95b	46.88 ± 6.02b	13.34 ± 1.51a	13.09 ± 3.79a	< 0.0001
luteolin	3.74 ± 0.49a	23.07 ± 6.61b	16.14 ± 1.23ab	5.64 ± 1.22ab	24.86 ± 4.76b	0.0078
myrycetin	4.46 ± 0.42a	9.52 ± 1.79a	3.09 ± 0.23a	8.48 ± 1.96a	8.29 ± 1.91a	N.S.
quercetin	40.59 ± 2.21c	11.23 ± 0.34a	8.25 ± 1.90a	22.74 ± 3.00b	8.56 ± 1.38a	< 0.0001

^1^ Means in rows followed by the same letter are not significantly different at the 5% level of probability *(p* < 0.05); ^2^ N.S. not significant statistically.

**Table 4 antioxidants-08-00458-t004:** The content of carotenoids (mg 100 g^−1^ FW) and chlorophylls (mg 100 g^−1^ FW) in examined raspberry leaves depending on cultivation system. Data are presented as the mean ± SE with ANOVA *p*-value.

Examined Compounds	Organic Raspberry (*n* = 20)	Conventional Raspberry (*n* = 24)	*p*-Value
**total carotenoids**	2.61 ± 0.12b ^1^	3.14 ± 0.10a	0.0014
neoxanthin	0.045 ± 0.01a	0.025 ± 0.00b	0.0013
lutein	1.23 ± 0.05a	1.06 ± 0.03b	0.0069
zeaxanthin	0.80 ± 0.03a	0.70 ± 0.02b	0.0118
violaxanthin	0.017 ± 0.001b	0.026 ± 0.002a	0.0004
alpha-carotene	0.060 ± 0.01b	0.109 ± 0.01a	0.0001
beta-carotene	0.46 ± 0.03b	1.22 ± 0.06a	< 0.0001
total chlorophylls	5.75 ± 0.30b	10.52 ± 0.60a	< 0.0001
chlorophyll b	1.79 ± 0.08b	2.43 ± 0.12a	0.0001
chlorophyll a	3.96 ± 0.23b	8.09 ± 0.48a	< 0.0001
chlorophyll a/b	2.19 ± 0.06b	3.29 ± 0.05a	< 0.0001

^1^ Means in rows followed by the same letter are not significantly different at the 5% level of probability (*p* < 0.05).

**Table 5 antioxidants-08-00458-t005:** The content of carotenoids (mg 100 g^−1^ FW) and chlorophylls (mg 100 g^−1^ FW) in examined raspberry leaves depending on cultivar. Data are presented as the mean ± SE with ANOVA *p*-value.

Examined Compounds	‘Polana’ cv. (*n* = 8)	‘Polka’ cv. (*n* = 12)	‘Tulameen’ cv. (*n* = 8)	‘Laszka’ cv. (*n* = 8)	‘Glen Ample’ cv. (*n* = 8)	*p*-Value
total carotenoids	2.70 ^1^ ± 0.07a	3.00 ± 0.22a	2.88 ± 0.19a	3.17 ± 0.12a	2.72 ± 0.22a	N.S. ^2^
neoxanthin	0.033 ± 0.00a	0.024 ± 0.00a	0.037 ± 0.01a	0.062 ± 0.01b	0.021 ± 0.00a	< 0.0001
lutein	1.08 ± 0.03a	1.15 ± 0.04a	1.23 ± 0.12a	1.19 ± 0.08a	1.05 ± 0.03a	N.S.
zeaxanthin	0.70 ± 0.02a	0.75 ± 0.03a	0.80 ± 0.08a	0.79 ± 0.05a	0.68 ± 0.02a	N.S.
violaxanthin	0.024 ± 0.002a	0.024 ± 0.002a	0.018 ± 0.001a	0.017 ± 0.001a	0.025 ± 0.004a	N.S.
alpha-carotene	0.070 ± 0.01a	0.088 ± 0.01a	0.100 ± 0.01a	0.095 ± 0.02a	0.080 ± 0.02a	N.S.
beta-carotene	0.79 ± 0.09a	0.96 ± 0.14a	0.70 ± 0.04a	1.01 ± 0.18a	0.87 ± 0.20a	N.S.
total chlorophylls	7.64 ± 0.35a	9.13 ± 1.06a	6.73 ± 0.30a	9.62 ± 1.44a	8.25 ± 1.49a	N.S.
chlorophyll b	1.95 ± 0.04a	2.23 ± 0.19a	1.93 ± 0.14a	2.45 ± 0.21a	2.10 ± 0.26a	N.S.
chlorophyll a	5.69 ± 0.31a	6.90 ± 0.87a	4.81 ± 0.17a	7.17 ± 1.23a	6.15 ± 1.24a	N.S.
chlorophyll a/b	2.91 ± 0.11a	2.93 ± 0.17a	2.56 ± 0.11a	2.75 ± 0.27a	2.71 ± 0.27a	N.S.

^1^ Means in rows followed by the same letter are not significantly different at the 5% level of probability (*p* < 0.05); ^2^ N.S. not significant statistically.

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
