# Peer review of "Phenolics and Carotenoid Contents in the Leaves of Different Organic and Conventional Raspberry (Rubus idaeus L.) Cultivars and Their In Vitro Activity"

_antioxidants, 2019, doi:10.3390/antiox8100458_

Round 1

Reviewer 1 Report

The authors submitted an interesting manuscript, which deals with analyses of polyphenolic, chlorophyll and carotenoid contents of Rubus idaeus leaves. The leaves are often parts of various tea mixtures. The authors analysed the leaves of different cultivars of this plant species. 

The authors used adequate methods. The plant material was prepared and extracted properly. Furthermore, the parameters of the used analytical methods were optimal. 

The antioxidant effects were evaluated photometrically by scavenging of ABTS  radical. This method is very popular for antioxidant activities screening.  For such exploration, this method was adequate.

The data support the results, which are well discussed. Furthermore, the authors cite the most important references.

The manuscript present a good quality phytoanalytical work. Therefore, I recommend the manuscript for publication.

Author Response

Replies to Reviewer no. 1 comments

Ms. Ref. No.:  antioxidants-594529, Title: “Phenolics and carotenoid contents in the leaves of different organic and conventional raspberry (Rubus idaeus L.) cultivars and their in vitro activity” Journal: Antioxidants

Dear Reviewer,

Thank you for giving me some comments/suggestions/opinion to prepared manuscript. In separated file Authors give all replies for your comments/suggestions/opinion

Reviewer 2 Report

According to the authors the manuscript entitled “Phenolic and carotenoids contents in the leaves of the different organic and conventional raspberry (Rubus idaeus L.) cultivars and their in vitro activity” evaluated the impact of organic and conventional farm management and harvest time on the polyphenol, carotenoids, chlorophyll and macroelement content in different raspberry cultivars. In my opinion, this is not true, since according to the Plants origin section they were collected at the time of cultivation, so how is possible to study harvest time? Moreover, for this purpose more sampling at different periods should be performed. The authors should rewritte the aims of this study.

A special attention should be given to formation, since sometimes the authors used mL other ml, sometimes the references appear [19] another 19, sometimes appear figures other fig. All abbreviations should be described when mentioned for the first time. The authors should pay attention to significant numbers.

Abstract is too confusing, the authors should rewritte and this section should contain the main purpose of the study, results and main conclusions, not the information related to sampling.

Discussion: The authors should add references to support the information in lines 205-209.

Table 1….cultivation in (2013-2014)? According abstract and material and methods section, the experimental was carried out in 2013.

Tables 2 to 4. The information in point 1 should be added in title of tables, instead of notes. Point 2. The authors must use either uppercase or lowercase letters, not both types at the same time, is too confuse. In my opinion, the authors should present the total content in a table, and the individual content in another.

Figure 5. The y axis should range from 0 to 100

Figures 6 and 7. In my opinion, the authors have few samples to make corrections. Moreover, the axes must be adjusted to the data, as example the x axis should range from 40 to 80 or 100, as appropriate.

Author Response

Replies to Reviewer no. 2 comments

Ms. Ref. No.:  antioxidants-594529, Title: “Phenolics and carotenoid contents in the leaves of different organic and conventional raspberry (Rubus idaeus L.) cultivars and their in vitro activity” Journal: Antioxidants

Dear Reviewer,

Thank you for giving me some comments/suggestions/opinion to prepared manuscript. In separated file Authors give all replies for your comments/suggestions/opinion

Reviewer 3 Report

The manuscript entitled "Phenolics and carotenoid contents in the leaves of different organic and conventional raspberry (Rubus idaeus L.) cultivars and their in vitro activity” addresses some important questions regarding to obtain new sources of natural antioxidants. Authors determine the polyphenol, carotenoid and chlorophyll profile in leaves of selected raspberry cultivars and analyze the effect of farm management on their content in different raspberry leaves cultivars. Some sections are so confusing and they need to be clarified. Some points are:

-Page 3 –lines 104-133. Section 2.6 should be reworded. The title of section 2.6 could be as “ABTS·+ radical cation scavenging assay” and a new subsection “antioxidant activity measurement” could be added with lines 118-120 and 130-133. Besides, authors should explain how the scavenging activity is determined (e.g. used equation) and explain that results are expressed in terms of trolox equivalents. In line 120, “….3mL of the PBS solution in PBS” should be ““….3mL of the ABTS·+ solution in PBS”.  Preparation of PBS is so confusing. Authors should reword.

-Page 5 – lines 186-188. Please, check. Are “Laszka” and “Glen Ample” the cultivars with the strongest antioxidant activity?

-Page 5 – lines 201-202. Please check. “Glen Ample” cv or “Laszka”?

-Page 5 – line 206. Some references about “health benefits” should be included.

-Page 5 –line 211 and others lines. References are not given in the same style. For example, compare the style for references in the line 211 and in the line 217. Revision of the style references in whole manuscript is recommended.

-Page 6 – lines 259-264- Full name for TE, DW and FW should be indicated for the first time.

- Page 6 – line 264.  Check subscripts and superscripts.

- Page 6 – line 266. Could  the R2 value = 0.83 be considered  as a “strong correlation”? More details on the level of errors in the measurements should be included. Authors should include the statistical errors on the Figures 6-7. Some results in the figures 6-7 show a poor linear fit. Why? The scale of the y-axis for Polana is so high on the plot. It suggests that the antioxidant activity does not change with the total polyphenols content. What is the meaning of the numbers “44, 77, 70, 95, 13 on the plots?. For Laszka cv. (figure 7), more points between 60-100 of the total polyphenols (mg/100 g-1 FW) could be necessary.

Author Response

Replies to Reviewer no. 3 comments

Ms. Ref. No.:  antioxidants-594529, Title: “Phenolics and carotenoid contents in the leaves of different organic and conventional raspberry (Rubus idaeus L.) cultivars and their in vitro activity” Journal: Antioxidants

Dear Reviewer,

Thank you for giving me some comments/suggestions/opinion to prepared manuscript. In separated file Authors give all replies for your comments/suggestions/opinion

Reviewer 4 Report

In Abstract, the results of antioxidant activity must be reported.

The effect of agrotechnology (organic versus conventional cultivation) on bioactive compounds and plant antioxidant potential  must be described/discussed  in the Introduction and in Discussion.

L. 28 – Too trivial sentence!

L. 31 – It should be “by-products”>

L. 36 and other places – “-O-“ must be in italic.

L37, 40, 41 – It must be [7,8} …. [11,12] … [13,14].

L. 43 – It must be “[15-18].

L. 104 – It should be ABTS radical cation preparation.

L. 107 – It must be “ABTS” not “ABTS●+”.

L. 110-120 – Remove this part.

L. 128 – Centrifugation must be characterized by “x g” instaed of “rpm”.

L. 134 -142 ; 194 – 203 –Macroelement content is not related to the subject of this paper and must be omitted.

Figure 5 – Results of the statistical analysis must be added!

Author Response

Replies to Reviewer no. 4 comments

Ms. Ref. No.:  antioxidants-594529, Title: “Phenolics and carotenoid contents in the leaves of different organic and conventional raspberry (Rubus idaeus L.) cultivars and their in vitro activity” Journal: Antioxidants

Dear Reviewer,

Thank you for giving me some comments/suggestions/opinion to prepared manuscript. In separated file Authors give all replies for your comments/suggestions/opinion

Round 2

Reviewer 2 Report

The manuscript entitled “Phenolics and carotenoid contents in the leaves of different organic and conventional raspberry (Rubus 4 idaeus L.) cultivars and their in vitro activity” should be accepted in the current form, since this new version showed a significant quality improvement.

Author Response

Review no. 2:

Thank you very much for the review (Round no. 2) and for your positive recommendation to publish our manuscript in the ‘Antioxidants’ journal.

Reviewer 3 Report

Thank you for your revised manuscript. Reviewer is pleased to see that authors have taken of the majority of the reviewers' comments. The manuscript would be publishable.

Author Response

Review no. 3:

Thank you very much for the review (round no. 2) and for your positive recommendation to publish our manuscript in the ‘Antioxidants’ journal.

As Reviewer pointed in their opinion: “English language and style are fine/minor spell check required.”

Authors’ response: the whole manuscript has been thoroughly checked by a professional translation company. The authors attach the appropriate language certificate.

Reviewer 4 Report

The authors corrected this paper properly taken under considerations all my comments. Therefore, I can accept it now.

Author Response

Review no. 4:

Thank you very much for the review (round no. 2) and for your positive recommendation to publish our manuscript in the ‘Antioxidants’ journal.

As Reviewer pointed in their opinion: “English language and style are fine/minor spell check required.”

Authors’ response: the whole manuscript has been thoroughly checked by a professional translation company. The authors attach the appropriate language certificate.
